# Free Fatty Acids Signature in Human Intestinal Disorders: Significant Association between Butyric Acid and Celiac Disease

**DOI:** 10.3390/nu13030742

**Published:** 2021-02-26

**Authors:** Simone Baldi, Marta Menicatti, Giulia Nannini, Elena Niccolai, Edda Russo, Federica Ricci, Marco Pallecchi, Francesca Romano, Matteo Pedone, Giovanni Poli, Daniela Renzi, Antonio Taddei, Antonino S. Calabrò, Francesco C. Stingo, Gianluca Bartolucci, Amedeo Amedei

**Affiliations:** 1Department of Experimental and Clinical Medicine, University of Florence, 50134 Florence, Italy; simone.baldi@unifi.it (S.B.); giulia.nannini@unifi.it (G.N.); elena.niccolai@unifi.it (E.N.); edda.russo@unifi.it (E.R.); antonio.taddei@unifi.it (A.T.); 2Department of Neurosciences, Psychology, Drug Research and Child Health Section of Pharmaceutical and Nutraceutical Sciences University of Florence, 50139 Florence, Italy; marta.menicatti@unifi.it (M.M.); marco.pallecchi@unifi.it (M.P.); gianluca.bartolucci@unifi.it (G.B.); 3Department of Biomedical, Experimental and Clinical Sciences “Mario Serio” University of Florence, 50134 Florence, Italy; f.ricci@unifi.it (F.R.); daniela.renzi@unifi.it (D.R.); antoninosalvatore.calabro@unifi.it (A.S.C.); 4Laboratory of Clinical Pathology, Versilia Hospital, 55041 Lido di Camaiore, Italy; francesca.romano@uslnordovest.toscana.it; 5Department of Statistics, Computer Science, Applications “G. Parenti”, 50134 Florence, Italy; matteo.pedone@unifi.it (M.P.); giovanni.poli@stud.unifi.it (G.P.); francescoclaudio.stingo@unifi.it (F.C.S.); 6SOD of Interdisciplinary Internal Medicine, Azienda Ospedaliera Universitaria Careggi (AOUC), 50134 Florence, Italy

**Keywords:** free fatty acids, butyric acid, GC–MS method, celiac disease, colorectal cancer

## Abstract

Altered circulating levels of free fatty acids (FFAs), namely short chain fatty acids (SCFAs), medium chain fatty acids (MCFAs), and long chain fatty acids (LCFAs), are associated with metabolic, gastrointestinal, and malignant diseases. Hence, we compared the serum FFA profile of patients with celiac disease (CD), adenomatous polyposis (AP), and colorectal cancer (CRC) to healthy controls (HC). We enrolled 44 patients (19 CRC, 9 AP, 16 CD) and 16 HC. We performed a quantitative FFA evaluation with the gas chromatography–mass spectrometry method (GC–MS), and we performed Dirichlet-multinomial regression in order to highlight disease-specific FFA signature. HC showed a different composition of FFAs than CRC, AP, and CD patients. Furthermore, the partial least squares discriminant analysis (PLS-DA) confirmed perfect overlap between the CRC and AP patients and separation of HC from the diseased groups. The Dirichlet-multinomial regression identified only strong positive association between CD and butyric acid. Moreover, CD patients showed significant interactions with age, BMI, and gender. In addition, among patients with the same age and BMI, being male compared to being female implies a decrease of the CD effect on the (log) prevalence of butyric acid in FFA composition. Our data support GC–MS as a suitable method for the concurrent analysis of circulating SCFAs, MCFAs, and LCFAs in different gastrointestinal diseases. Furthermore, and notably, we suggest for the first time that butyric acid could represent a potential biomarker for CD screening.

## 1. Introduction

Fatty acids are carboxylic acids classified by the length of their aliphatic chain, the presence or absence of double bonds, and the location of double bonds. Short chain fatty acids (SCFAs) have five carbon atoms or fewer, medium chain fatty acids (MCFAs) have from six to 12 carbon atoms, while long chain fatty acids (LCFAs) have 13–20 carbon atoms [1].

Although fatty acids are usually considered a simple energy source, in recent decades they have emerged as fundamental intracellular and extracellular signaling molecules. They: (i) modulate the activation of gene transcription, (ii) regulate the post-transcriptional modification of proteins, and (iii) act as coactivators of enzymes [2,3].

A main source of lipids is plasmatic free fatty acids (FFAs), namely SCFAs, MCFAs, and LCFAs, which are hydrolyzed from adipose tissue stores and carried by circulating albumin to provide energy for tissues during fasting [4].

In particular, SCFAs are the main metabolites produced by the bacterial anerobic fermentation of indigestible polysaccharides and proteins in the large intestine. They are absorbed by colonocytes or transported into the portal circulation to be metabolized by hepatocytes, while the remaining SCFAs enter systemic circulation [5,6].

In general, they contribute to regulate the glucose and cholesterol metabolisms to maintain the intestinal barrier integrity and to modulate the differentiation of T cells [7,8].

The adequate gut microbiota production of SCFAs is critical to maintaining the host’s normal gut physiology and metabolic functions and dietary fiber [9]. Probiotic supplementation [10] can increase the capacity to produce larger quantities of SCFAs, with benefits for the host health.

Differently from SCFAs, MCFAs and LCFAs are generally encountered in the diet, especially from milk and dairy products, and are important regulators of energy metabolism, gene expression, ion channels and pump activities, membrane trafficking, and immune processes [11,12,13].

It is well known that higher FFA concentration causes an increase of oxidative stress and exerts pro-inflammatory effects. In particular, altered levels of FFAs have been associated with Crohn’s disease, autoimmune disorders, and various types of cancer [14,15,16,17].

Hence, as FFAs play important roles for the host, and their imbalance is linked to multiple metabolic, gastrointestinal, and malignant diseases, investigation into different biological samples is becoming noteworthy with the aim of researching new metabolic biomarkers [18,19,20]. However, it is important to underline that altered lipid profiles can be caused by dysregulated bacterial fermentation, an unbalanced diet, and de novo synthesis in cancer tissues [21].

Finally, concerning the serum profiles of FFAs, a quantitative determination with gas chromatography–mass spectrometry (GC–MS) was recently proposed, with advantages of high sensitivity, peak resolution, and reproducibility.

Starting from these premises and using Dirichlet-multinomial regression, in this explorative study we compared the serum FFA profile of patients with different gastrointestinal diseases, including celiac disease (CD), adenomatous polyposis (AP), and colorectal cancer (CRC), to healthy controls (HC) in order to highlight disease-specific FFA signature.

## 2. Materials and Methods

### 2.1. Study Design and Patient Enrolment

We used a dedicated previously described study protocol [22]. Briefly, 44 patients affected by different gut diseases (19 CRC, 9 AP, 16 CD) and 16 healthy controls (HC) were enrolled in different studies between January 2016 and February 2019 at the Azienda Ospedaliera Universitaria Careggi, Italy.

At enrollment, people with CRC, AP, and CD showed a median BMI < 25, all patients were on omnivorous diets, and none of them reported special dietary habits or dietary restrictions.

All CRC patients were affected by nonmetastatic colon cancer, and in particular 16 were at stage I-II, and three were at stage III.

Table 1 reports clinical characteristics of patients.

Each patient’s whole blood was collected in a clinical tube containing 3.2% sodium citrate as anticoagulant and then centrifuged at 1500× *g* for 10 min, then the serum was collected and stored at −20 °C until analysis.

The study received approval from the local ethics committee—Comitato Etico Regionale per la Sperimentazione Clinica della Regione Toscana, Sezione AREA VASTA CENTRO Institutional Review Board (CE: 11166_spe, 11/09/2018 and CE: 10443_oss, 14/02/2017)—and informed written consent was obtained from each participant.

### 2.2. FFAs Determination by GC–MS Analysis

The FFA analysis was performed using an Agilent GC–MS system composed of a 5971 single quadrupole mass spectrometer, a 5890 gas-chromatograph, and a 7673 autosampler. The chemicals, GC–MS conditions, and calibration parameters were reported in Appendix A.

### 2.3. Sample Preparation

Just before the analysis, each sample was thawed. The FFAs were extracted as follows: an aliquot of 300 µL of plasma sample was added to 10 μL of ISTD mixture, 100 μL of tert-butyl methyl ether, and 20 µL of 6 M HCl + 0.5 M NaCl solution in 0.5 mL centrifuge tube. Afterwards, each tube was stirred in a vortex for 2 min, centrifuged at 10,000 rpm for 5 min, and finally the solvent layer was transferred in a vial with microvolume insert and analyzed (see Appendix A).

### 2.4. Statistical Analysis

Statistical analysis of the FFA percentages was performed in R (R Core Team, version 3.5.3, Wien, Austria), and all the graphs were plotted with ggplot2 (R Core Team, version 3.1.1, Wien, Austria). Pairwise comparisons of FFA composition between patient groups were assessed using the Wilcoxon rank-sum test.

In order to highlight the separation between groups taking into account the diverse FFA percentage compositions, partial least squares discriminant analysis (PLS-DA) was performed with R package “DiscrMiner” (R Core Team, version 0.1-29, Wien, Austria).

*p*-values of less than 0.05 were considered statically significant, no multiplicity correction was applied, and findings were interpreted as hypothesis generating.

A Bayesian Dirichlet-multinomial regression model was implemented to explore the associations between clinical variables and serum FFA composition. Relevant associations were selected using our Bayesian variable selection (BVS) method [23]. This method detects both main effects and interactions between explanatory variables. This means that if the association of one clinical variable with serum FFA composition depends on the state of another clinical variable, the model effectively takes into account these relationships. The method’s output is a list of posterior probability of inclusion (PPI) and the posterior mean of the non-zero regression coefficients. PPI is the probability, between 0 and 1, that a given association is non-zero, accounting for the effect of all other clinical variables. The posterior mean is an estimate of the effect size of a given association.

## 3. Results 

### 3.1. Descriptive Analysis of Serum FFA Distributions

Explorative descriptive analysis was firstly performed on the marginal distributions of each serum FFA percentage (Table 2).

As reported in Figure 1, healthy controls showed an evidently different composition of serum FFAs than CRC, AP, and CD patients, with a high percentage of acetic acid and a reduction of other FFA levels such as isobutyric, isovaleric, 2-methylbutyric, heptanoic, dodecanoic, tetradecanoic, hexadecanoic, and octadecanoic acids.

In particular, the quantitative analysis of MCFAs was conducted taking into account the isovaleric acid ISTD, while the LCFA quantification was performed with a semiquantitative analysis using the nonanoic acid as reference. It is important to point out that abundance of valeric acid was often below the minimum detectable value; although these values do not have an impact on other abundances, all statistics and tests concerning valeric acid need to be carefully interpreted.

Figure 2 reports in detail each FFA level for CRC, CD, AP patients and HC, while the *p*-values of the pairwise comparisons conducted for FFA percentages are shown in Table 3.

In detail, compared to CD patients, HC showed a significantly higher quantity of acetic, propionic, valeric, octanoic, and nonanoic acids and a significantly lower percentage of butyric, 2-methylbutyric, isobutyric, isovaleric, hexanoic, and decanoic acids and LCFAs.

CRC patients displayed a significantly lower percentage of acetic, propionic, butyric, valeric, and nonanoic acids and significantly higher percentages of 2-methylbutyric, isobutyric, isovaleric, hexanoic, and heptanoic acids and LCFAs compared to HC.

Compared to AP patients, HC displayed a significantly higher percentage of acetic, propionic, butyric, valeric, heptanoic, and nonanoic acids and a significant reduction of butyric, 2-methylbutyric, isovaleric, hexanoic, tetradecanoic, hexadecanoic, and octadecanoic acid abundances.

In addition, compared to CRC patients, CD patients showed significantly higher percentages of propionic, butyric, valeric, heptanoic nonanoic, dodecanoic, and tetradecanoic acids; however, they reported significantly reduced levels of isobutyric and isovaleric acids.

Otherwise, CD patients showed significantly higher percentages of propionic, isobutyric, butyric, valeric, heptanoic, nonanoic, dodecanoic, and tetradecanoic acids compared to AP patients, while lower levels of isovaleric and hexanoic acids are noted.

To conclude, CRC patients displayed a high percentage of only hexanoic acid in comparison to AP patients.

Furthermore, to characterize a state-specific FFA profile, we performed PLS-DA on the FFA percentage matrix. The classifier confirmed a perfect overlap between the CRC and AP patients and a separation of HC from the diseased groups (Figure 3).

As described in Table 4, AP patients are generally classified as CRC (8/9), most HC are correctly classified (15/16), and none of the misclassified subjects were assigned to this group. Most classification errors involved the CD state. Indeed, one AP and two CRC subjects were classified as CD, while two CD were assigned to CRC status. This suggests that CD subjects have an FFA composition closer to unhealthy subjects than controls.

### 3.2. Dirichlet-Multinomial Regression

Dirichlet-multinomial regression was performed to identify variables, and their interactions, that have an effect on the serum FFA composition.

The following clinical covariates were included in the analysis: age, BMI, gender, and group. Group is a categorical variable, and its categories are CD, AP, and CRC. The baseline category for gender is female, and in order to highlight any differences between inflammatory intestinal diseases, the baseline category for group is AP.

This approach identified a positive association between CD and butyric acid; this association is of a large magnitude (the posterior mean is equal to 3.8548), and it is strongly supported by the data (PPI = 1.0). All other associations were not included as they were not supported by the data. All other PPIs were less than 0.1. Among patients of the same age with the same BMI and same gender, those with CD, compared to those with AP, have on average a stronger prevalence of butyric acid in their FFA composition.

We also evaluated possible interaction effects. Results are reported in Table 5: CD shows significant interactions with age (PPI = 0.6676), BMI (PPI = 0.8411), and gender (PPI = 0.7380).

The magnitude of these interactions is moderate, and it should be interpreted as “deviation” from the CD–butyric acid effect. That means, e.g., that on average among patients with the same age and same gender, every additional BMI point is associated with a 0.1161 increase of the CD effect on the (log) prevalence of butyric acid in FFA composition. Further interesting information given by the interaction is that on average among patients with the same age and same BMI, being male compared to being female implies a 0.2278 decrease of the effect of CD on the (log) prevalence of butyric acid in FFA composition. The 3D plots in Figure 4 illustrate the prevalence of butyric acid as a function of age and BMI in a female subject affected or not affected by CD.

## 4. Discussion

It is well documented that circulating FFAs act as signaling molecules, are fundamental components of cellular structures, and are important sources of energy [24], and many studies demonstrated the association between altered FFA levels and different pathological conditions, such as inflammatory and cardiovascular diseases, neurological disorders, and cancer [25,26,27,28].

Thus, the quantitative and qualitative analysis of FFAs in various biological specimens, like stool, urine, saliva, and blood, has recently attracted attention in efforts to identify new potential biomarkers [29,30,31,32].

In recent years, different analytical technologies have been used to detect FFAs, including gas chromatography–mass spectrometry, gas chromatography with flame ionization detection (GC–FID), and liquid chromatography–mass spectrometry (LC-MS) [33,34,35]. However, GC–MS is currently the most frequently used method for FFA analysis because it provides higher selectivity, specificity, and accuracy compared to other detection methods [36].

In the present study, we therefore used for the first time a GC–MS method to perform a simultaneously qualitative and quantitative analysis of SCFAs, MCFAs, and LCFAs in serum samples of healthy controls and patients with different intestinal diseases, namely adenomatous polyposis, colorectal cancer, and celiac disease.

First, the PLS-DA model used for the FFA percentage matrix showed a clear separation between HC and the other three disease groups, a perfect overlap between CRC and AP patients, and a partial separation between CD patients and the two coincident groups of CRC and AP patients.

Recently, using the same GC–MS system, we analyzed the fecal SCFA profile of the same patients and found a definite overlap between HC and CD patients, while CRC and AP patients displayed a divergence from the other two classes and did not completely coincide [22].

So, taking into account only the SCFA abundances, we did not find a relationship between circulating and fecal concentration. In agreement with our results, Muller et al. reported that fecal acetic acid and butyric acid were not related to their respective plasmatic concentrations, and only propionic acid seems to be reflective of its respective fecal concentration [37].

Nevertheless, both CRC and AP patients showed similar profiles of serum FFAs and, as we previously demonstrated [22], of fecal SCFAs, so these results strengthen the established evidence that adenomas can evolve into cancers following the adenoma–carcinoma sequence.

Moreover, the healthy controls displayed different FFA profiles compared to AP, CRC, and CD patients. In particular, CRC and AP patients showed lower levels of SCFAs compared to HC, except for 2-methylbutyric acid, isobutyric acid, and isovaleric acid, which were more abundant in HC.

In accordance with our results, Yusuf et al. reported that CRC patients had lower plasma levels of acetic, propionic, and butyric acids than healthy subjects; on the contrary, Amiot et al. reported that acetate, propionate, and butyrate were increased in fecal samples of CRC patients [38,39].

SCFA-producing bacteria abundance is closely negatively related to the degree of malignancy, and it has recently emerged that the manipulation of intestinal SCFA levels could be a preventive/therapeutic strategy for CRC because of their anti-inflammatory and anti-tumorigenesis properties [40,41].

MCFAs are potent agonists of peroxisome proliferator-activated receptors and are involved in cell death and survival regulation [42,43]. CRC and AP patients showed elevated percentages of hexanoic, heptanoic, octanoic, decanoic, and dodecanoic acids compared to HC, while nonanoic acid was higher in HC.

However, contrary to a study conducted on 117 plasma samples of CRC patients (at different tumor stages) [44], which reported higher levels of hexanoic acid in high-grade dysplasia adenoma patients compared to CRC patients, we found that AP patients showed lower levels of hexanoic acid than people with CRC. These different findings could be explained by the fact that our enrolled AP patients showed low-grade dysplasia.

In according to our results, Crotti et al. found that both CRC and high-grade dysplasia adenoma patients showed significantly higher levels of octanoic and dodecanoic acids. Moreover, they also reported higher levels of decanoic acid and suggest its potential role as a specific CRC biomarker [44].

In addition, Iemoto et al. documented that the serum level of octanoic acid was significantly associated with disease progression, so it may serve as a useful predictor for CRC prognosis [45].

Compared to HC, CRC and AP patients showed higher abundances of LCFAs. Similar to MCFAs, LCFAs play key roles in cell homeostasis and the maintenance of cellular integrity and act as activators of peroxisome proliferator-activated receptor [46].

Different studies reported that an increased concentration of LCFAs and very long fatty acids was associated with an increased CRC risk, for example, linoleic acid and palmitic acid enhance colon carcinogenesis and metastasis [47,48,49,50].

Compared to HC, CD patients showed lower levels of acetic, propionic, and valeric acids but higher percentages of other SCFAs (butyric, 2-methylbutyric, isobutyric, and isovaleric acids) and MCFAs and LCFAs. On the contrary, Jakobsdottir et al. found that both CD patients and healthy subjects showed similar levels of serum SCFAs, while many papers reported that fecal SCFAs could be a signature for CD [51,52,53].

Since fecal SCFA production is related to the composition of gut microbiota and the availability of fermentable substrates, is well documented that CD patients with a reduction of nutrient absorption, caused by villous atrophy, show lower abundances of fecal SCFAs [54]. Instead, MCFAs have been identified as discriminatory metabolites for intestinal bowel disease (IBD), showing significantly decreased results in IBD patients compared to HC. However, in accordance with our data, Solakivi et al. found that CD patients displayed higher levels of serum LCFAs than HC [55,56].

We used a Dirichlet-multinomial regression model with the aim of evaluating which variables and their interactions had a significant effect on the serum FFA profiles.

Interestingly, we found a strong correlation between CD and butyric acid, and we reported interaction effects between the age, BMI, and gender variables and CD.

Butyric acid, one of the most abundant SCFAs in the human colon, is quickly consumed by the intestinal epithelium and used by colonocites primarily as an energy source.

Butyric acid is also an essential regulator of intestinal homeostasis and notably is involved in the processes of anti-inflammation and immunomodulation, enhancing the naïve T-cell polarization to Tregs or Th17 and Th1 effector cells [8,57].

Although systemic butyrate appears to limit the antitumor effect of the anti-CTLA-4 blockade [58], the importance of butyric acid, in particular for human colon health, has been documented in many studies conducted on patients with inflammatory gut diseases. For example, butyrate has been hypothesized to play a key role in reducing hypersensitivity to intestinal receptors, so its supplementation seems to be a promising therapy for irritable bowel syndrome [59].

Regarding CD, it is well established that besides environmental and genetic factors, an imbalance of the microbiota composition is related to its onset and consequent altered intestinal metabolic profile, including SCFA production [52,60]. Indeed, some authors reported that fecal SCFA amounts were higher in patients with untreated and treated CD than healthy adults [53,61]. Therefore, due to the altered gut homeostasis, microbial metabolites such as butyric acid could easily enters the systemic circulation of CD patients.

In addition, we found that only in female patients affected by CD did the prevalence of butyric acid rise with increasing BMI. While the higher CD prevalence in female compared to male individuals is well established, contrasting evidence has been found regarding the relationship between CD and BMI [62]. Many studies found a positive correlation between BMI and serum SCFAs, particularly propionic, butyric, and isovaleric acid in obese patients compared to lean individuals [63,64].

However, Males affected by CD are more commonly overweight than females [65], and CD females have significantly lower mean BMI than the general population [66].

Moreover, sex differences between males and females are responsible for a dissimilar flux of fatty acids, but in response to obesity, both men and women show increased fatty acid release into the bloodstream [67]. However, a different study reported contrasting data, documenting higher serum FFAs in women with respect to men with obesity [68].

Inflammation processes associated with both obesity and CD may lead to loss of intestinal integrity, known as “leaky gut syndrome”, which is responsible for an increased permeability of the intestinal mucosa that could allow microorganisms, small molecules, and bacterial metabolites like SCFAs to enter the bloodstream [69,70,71]. Finally, our different findings between males and females could be supported by a “hormonal hypothesis” because female reduced fertility, delayed menarche, amenorrhea, and early menopause often represent the initial clinical features that ultimately result in a CD diagnosis [72].

Thus, we propose that butyric acid could represent a potential biomarker for screening of CD, and more studies will be necessary in order to evaluate the serum levels of butyric acid in other intestinal pathologies. The serum FFA profile could potentially be used to discriminate CD patients from potential CD patients.

## 5. Conclusions

Despite our findings being exploratory for the limited sample size, we propose a GC–MS method for the concurrent analysis of SCFAs, MCFAs, and LCFAs in different gastrointestinal diseases. For the first time, we highlighted the existence of a serum SCFA signature in HC and in patients with CRC and AP, while the CD patients did not show a profile completely distinguishable from HC and CRC patients. However, we reported a strong interaction between butyric acid and CD. Of course, a better investigation of the circulating butyric acid in other intestinal diseases such as IBD is needed. Nevertheless, our results suggest that butyric acid could be a potential (and easy to assess) biomarker for celiac disease.

## Figures and Tables

**Figure 1 nutrients-13-00742-f001:**
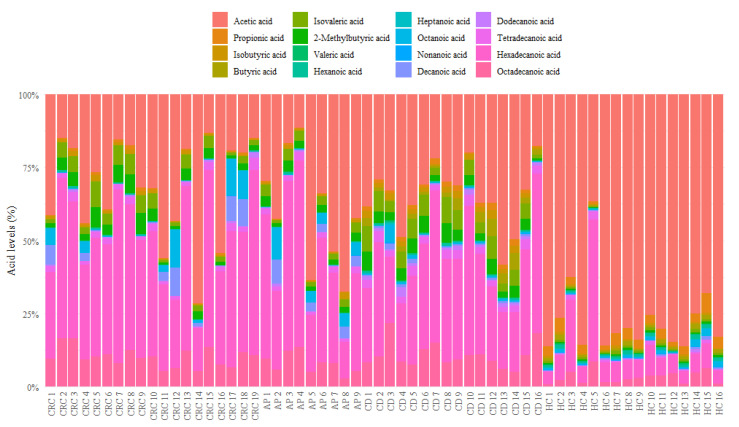
Bar plot of relative abundances of each serum FFA of CRC, AP, CD patients and healthy controls. FFA = free fatty acid; HC = healthy controls; CD = celiac disease; AP = adenomatous polyposis; CRC = colorectal cancer.

**Figure 2 nutrients-13-00742-f002:**
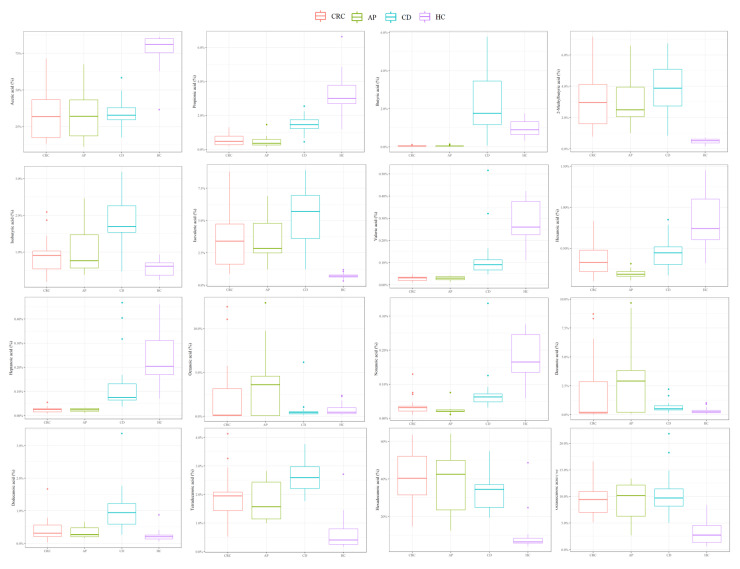
Boxplots representing each FFA percentage in CD, CRC, AP patients and HC.

**Figure 3 nutrients-13-00742-f003:**
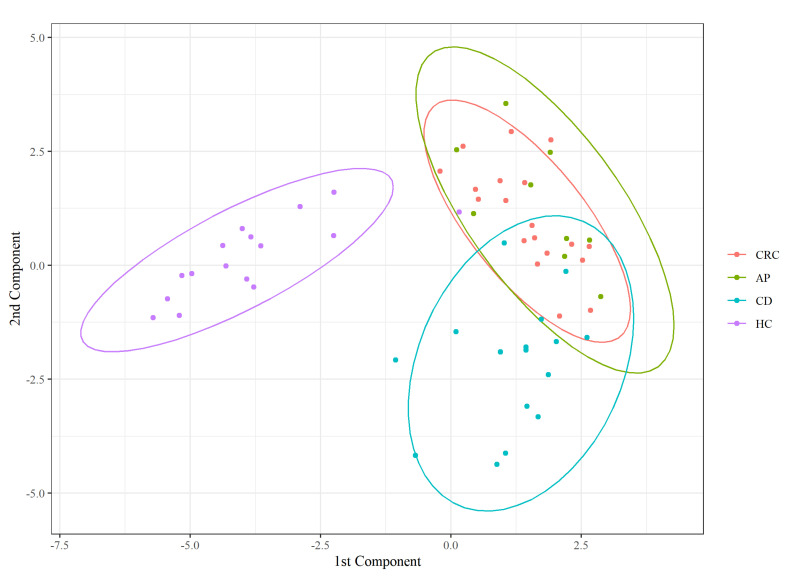
Partial least squares discriminant analysis (PLS − DA) score plot.

**Figure 4 nutrients-13-00742-f004:**
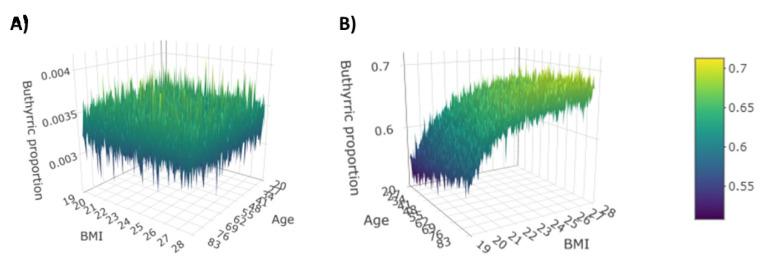
Prevalence of butyric acid for female without CD (**A**) and with CD (**B**).

**Table 1 nutrients-13-00742-t001:** Features of enrolled patients. BMI = body mass index; IQR = interquartile range. HC = healthy controls; CD = celiac disease; AP = adenomatous polyposis; CRC = colorectal cancer.

ID Patients	State	Number	Male/Female Ratio, n	Age, Median (IQR)	BMI, Median (IQR)
HC	Healthy control	16	1 (8/8)	41.7 (31.7)	NA
CD	Celiac disease	16	0.5 (6/10)	35.5 (21)	22.1 (3.7)
AP	Adenomatous polyposis	9	1.2 (5/4)	68 (28)	24.9 (3.9)
CRC	Colorectal cancer	19	8.5 (17/2)	80 (13.5)	24.5 (4.5)

**Table 2 nutrients-13-00742-t002:** Representation of the median (Q1–Q3) of each FFA percentage in CD, CRC, AP patients and healthy controls. FFA = free fatty acid; HC = healthy controls; CD = celiac disease; AP = adenomatous polyposis; CRC = colorectal cancer.

FFA	CRC	AP	CD	HC
Acetic acid	0.31768 (0.17476–0.43413)	0.3204 (0.18718–0.43255)	0.32748 (0.29762–0.37929)	0.81064 (0.75446–0.84983)
Propionic acid	0.00472 (0.00271–0.00774)	0.00356 (0.00256–0.00586)	0.01454 (0.01225–0.01744)	0.02999 (0.02685–0.03777)
Isobutyric acid	0.00909 (0.00546–0.01029)	0.00763 (0.00569–0.01471)	0.01695 (0.01537–0.0226)	0.00615 (0.00372–0.00706)
Butyric acid	0.00038 (0.00026–0.00058)	0.00042 (0.00035–0.00048)	0.01753 (0.01166–0.03449)	0.00897 (0.00639–0.01317)
Isovaleric acid	0.03399 (0.01612–0.04728)	0.02838 (0.02483–0.04788)	0.05703 (0.03606–0.06944)	0.00701 (0.00621–0.00793)
2–methylbutyric acid	0.02959 (0.01603–0.04119)	0.02493 (0.02058–0.0394)	0.03875 (0.02733–0.05085)	0.00517 (0.00375–0.00614)
Valeric acid	0.00032 (2e–04–0.00035)	0.00031 (0.00023–0.00038)	0.0009 (0.00068–0.00113)	0.0026 (0.00226–0.00376)
Hexanoic acid	0.00328 (0.00219–0.00477)	0.00185 (0.00161–0.00219)	0.00444 (0.00303–0.00517)	0.00741 (0.00606–0.01099)
Heptanoic acid	0.00025 (0.00016–0.00029)	0.00024 (0.00018–3e–04)	0.00075 (0.00064–0.00132)	0.00204 (0.0017–0.00311)
Octanoic acid	0.00137 (0.00086–0.03154)	0.03602 (0.00092–0.04563)	0.00409 (0.00278–0.00555)	0.00462 (0.00286–0.00977)
Nonanoic acid	0.00031 (0.00021–0.00035)	0.00021 (2e–04–0.00025)	0.00063 (0.00048–0.00072)	0.00165 (0.00135–0.00246)
Decanoic acid	0.00205 (0.00117–0.02861)	0.02912 (0.00203–0.03822)	0.00509 (0.00409–0.00773)	0.0024 (0.00165–0.0037)
Dodecanoic acid	0.0031 (0.0021–0.00558)	0.00266 (0.00202–0.00473)	0.00943 (0.00584–0.01224)	0.00206 (0.00136–0.00255)
Tetradecanoic acid	0.0195 (0.01432–0.02084)	0.01568 (0.0114–0.02433)	0.02588 (0.02206–0.02976)	0.00406 (0.00253–0.00795)
Hexadecanoic acid	0.40353 (0.31485–0.52113)	0.42495 (0.23516–0.49912)	0.34432 (0.24853–0.36942)	0.06569 (0.05869–0.08501)
Octadecanoic acid	0.09405 (0.06981–0.10899)	0.1013 (0.06261–0.12141)	0.09689 (0.08155–0.11434)	0.02715 (0.01355–0.04519)

**Table 3 nutrients-13-00742-t003:** *p*-values of the intergroup comparisons assessed with Wilcoxon tests conducted on FFA percentages. *p*-values less than 0.05 were considered statistically significant. FFA = free fatty acid; HC = healthy controls; CD = celiac disease; AP = adenomatous polyposis; CRC = colorectal cancer.

FFA	HC vs. CD	HC vs. CRC	HC vs. AP	CD vs. CRC	CD vs. AP	CRC vs. AP
Acetic acid	0.0000	0.0000	0.0000	0.7561	0.6368	0.9331
Propionic acid	0.0001	0.0000	0.0000	0.0000	0.0002	0.6291
Isobutyric acid	0.0000	0.0373	0.0741	0.0001	0.0096	0.6291
Butyric acid	0.0058	0.0000	0.0000	0.0000	0.0000	0.6993
Isovaleric acid	0.0000	0.0000	0.0000	0.0081	0.0272	0.7722
2 methylbutyric acid	0.0000	0.0000	0.0000	0.1613	0.2071	0.8470
Valeric acid	0.0001	0.0000	0.0000	0.0000	0.0000	0.7355
Hexanoic acid	0.0019	0.0002	0.0000	0.2300	0.0003	0.0284
Heptanoic acid	0.4909	0.0000	0.0000	0.0000	0.0000	0.9615
Octanoic acid	0.0001	0.1515	0.0671	0.1613	0.8028	0.3829
Nonanoic acid	0.0058	0.0000	0.0000	0.0003	0.0003	0.1423
Decanoic acid	0.0000	0.9870	0.2071	0.3011	0.2976	0.1716
Dodecanoic acid	0.0000	0.0263	0.0842	0.0002	0.0003	0.7355
Tetradecanoic acid	0.0000	0.0000	0.0014	0.0065	0.0080	0.4679
Hexadecanoic acid	0.0000	0.0000	0.0001	0.0948	0.3876	0.9231
Octadecanoic acid	0.0000	0.0000	0.0003	0.4814	0.6771	0.9615

**Table 4 nutrients-13-00742-t004:** Classification results of the PLS-DA model. PLS-DA = partial least squares discriminant analysis; HC = healthy controls; CD = celiac disease; AP = adenomatous polyposis; CRC = colorectal cancer.

		Predicted			
		CRC	AP	CD	HC
True status	CRC	17	0	2	0
	AP	8	0	1	0
	CD	2	0	14	0
	HC	1	0	0	15

**Table 5 nutrients-13-00742-t005:** Interaction effects evaluated with Dirichlet-multinomial regression between butyric acid, age, BMI, gender, CRC, and CD variables. Baseline category for gender is female and baseline category for state is AP. BMI = body mass index; HC = healthy controls; CD = celiac disease; AP = adenomatous polyposis; CRC = colorectal cancer.

Acid	Interaction	Gender	CRC	CD
	Age	-	-	0.0933
	BMI	-	-	0.1161
Butyric	Gender	-	-	−0.2278
	CRC	-	-	-
	CD	-	-	-

## Data Availability

Data is contained within the article.

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
