# Peer review of "Free Fatty Acids Signature in Human Intestinal Disorders: Significant Association between Butyric Acid and Celiac Disease"

_nutrients, 2021, doi:10.3390/nu13030742_

Round 1

Reviewer 1 Report

"Free fatty acids signature in human intestinal disorders: significant association between butyric acid and celiac disease" article explains the biomarker for intestinal inflammation which helps to identify with many diseases associated with intestine diseases. However, the authors use a very limited number of patients studied, and also lack many direct shreds of evidence.  Like, the authors talked about BMI, but not showing the real BMI correlation. In obesity, SCFA populations are very much affected and also obesity mediated CRC patients. Many research outlined, that supplementing SCFA mitigates CRC. The main resources of dietary intake of those patients' information are very well correlated with serum FFA. Those details are lack in this article. 

Authors' are requested to clarify those details which help the article more valuable. 

Author Response

Point 1: The authors use a very limited number of patients studied, and also lack many direct shreds of evidence.

Response 1: We agree with the reviewer suggestion, however this is an explorative and innovative study starting from our previous published work evaluating the fecal SCFA’ profiles on the same cohort of patients (doi: 10.3748/wjg.v25.i36.5543) which reported interesting results that we will confirm for the first time in serum. Of course, a future study with a higher number of enrolled patients focusing in a specific disease could help to better clarify the classification of these intestinal pathologies through the FFAs analysis.

Point 2: Like, the authors talked about BMI, but not showing the real BMI correlation. In obesity, SCFA populations are very much affected and also obesity mediated CRC patients. Many research outlined, that supplementing SCFA mitigates CRC. The main resources of dietary intake of those patients information are very well correlated with serum FFA. Those details are lack in this article.

Response 2: We thank the reviewer for appropriate suggestion raising the issue, and we agree with him for the need to clarify this point. Thus, we have added the information regarding patient’s diet and BMI in the method section (please see lines 87-89). However, on average, the enrolled patients showed normal weigh and followed a comparable diet. Of course, knowing their detailed dietary habits could be of interest but sorry questionnaires on eating habits were not collected. Finally, as reported in results section (line 213) we performed a regression analysis to evaluate the association between clinical variables and serum FFAs and we found no correlation between all patients and BMI. The probable explanation could be that obese subjects were missing in our patients’ cohort. Notably, a significant interaction between butyric acid levels and clinical parameters (BMI, age and sex) was found only in the group of CD patients.

Reviewer 2 Report

Reviewer's report

Title: Free fatty acids signature in human intestinal disorders: significant association between butyric acid and celiac disease

Date: 05.02.2021

Reviewer's report:

The content of the research "free fatty acids signature in human intestinal disorders" is of considerable interest; this is a professionally written manuscript. I have some minor comments to improve the document.

- Minor Essential Revisions

  • Table 1 line 2 (Celiac Disease) the male/female ratio seems odd, and in total, it looks like 18 patients instead of 16, maybe a typing mistake.
  • The same table CRC patients male/female ratio there is "," instead of "/" should be corrected
  • Line 87 I understand that clinical red top tube, but there are differences between the top colour of tubes in some countries. Therefore it would be better if they present it with the type of the tube, e.g. a standard tube, EDTA etc.
  • Lines 90-91: The full name of the Ethical Committee and the number of ethical approval should be presented.
  • Please add detailed explanations (abbreviations, statistical test used etc.) under all tables so that tables can be clearly read without going back to text.

Author Response

Point 1:  Table 1 line 2 (Celiac Disease) the male/female ratio seems odd, and in total, it looks like 18 patients instead of 16, maybe a typing mistake.

Response 1: Thank you for the right suggestion, we have correct the mistake.

Point 2:  The same table CRC patients male/female ratio there is "," instead of "/" should be corrected

Response 2: Thank you for the right suggestion, we have correct the mistake.

Point 3:  Line 87 I understand that clinical red top tube, but there are differences between the top colour of tubes in some countries. Therefore it would be better if they present it with the type of the tube, e.g. a standard tube, EDTA etc.

Response 3: We thank the reviewer for the appropriate observation, we have correct the mistake (please see line 97).

Point 4: Lines 90-91: The full name of the Ethical Committee and the number of ethical approval should be presented.

Response 4: In agreement with the reviewer suggestion, we have provided the full name of the Ethical Committee and the number of ethical approval (please see lines 100-103)

Point 5: Please add detailed explanations (abbreviations, statistical test used etc.) under all tables so that tables can be clearly read without going back to text

Response 5: As rightly suggested by the reviewer, we provide a detailed explanation under all tables.

Reviewer 3 Report

The study is interesting and original. The comparison of circulating fatty acids profile ( short, medium and long chain) considering intestinal pathologies is well argued. However, information is lacking  concerning  the type and extent of the CRC.

 In Section Introduction,the authors must add one  sentences about the potential benefial effects of  the  short fatty acids  as  metabolites of probiotics.

What about the comparison AP Vs CRC?  The authors should discuss this aspect.

Author Response

Point 1:  The study is interesting and original. The comparison of circulating fatty acids profile (short, medium and long chain) considering intestinal pathologies is well argued. However, information is lacking concerning the type and extent of the CRC.

Response 1: We thank the reviewer for the careful and focused analysis of our study. We add the requested information (see lines 90-91).

Point 2:  In Section Introduction, the authors must add one sentences about the potential benefial effects of the short fatty acids as metabolites of probiotics.

Response 2In agreement with the interesting suggestion of reviewer, we added and discussed this aspect in the introduction section (please see lines 56-59).

Point 3:  What about the comparison AP Vs CRC?  The authors should discuss this aspect.

Response 3:  We thank the reviewer for the appropriate suggestion and in agreement, we have discussed the differences among patients with AP and CRC in discussion section (please see lines 268-269 and 284-293.

Round 2

Reviewer 1 Report

The authors' responses to all the suggestions were raised in the review.